# Learning Multiple Coordinated Agents under Directed Acyclic Graph Constraints

## Abstract

This paper proposes a novel multi-agent reinforcement learning (MARL) method to learn multiple coordinated agents under directed acyclic graph (DAG) constraints. Unlike existing MARL approaches, our method explicitly exploits the DAG structure between agents to achieve more effective learning performance. Theoretically, we propose a novel surrogate value function based on a MARL model with synthetic rewards (MARLM-SR) and prove that it serves as a lower bound of the optimal value function. Computationally, we propose a practical training algorithm that exploits new notion of leader agent and reward generator and distributor agent to guide the decomposed follower agents to better explore the parameter space in environments with DAG constraints. Empirically, we exploit four DAG environments including a real-world scheduling for one of Intel's high volume packaging and test factory to benchmark our methods and show it outperforms the other non-DAG approaches.

## 1 Introduction

Multi-agent reinforcement learning (MARL) coordinates multiple subtasks to collaboratively achieve an optimal team reward as a shared goal (Zhang et al., 2021). However, most existing works do not generalize to the settings where multiple subtasks have a complex relationship where higher-level subtasks are affected by lower-level subtasks (Yang et al., 2020; Foerster et al., 2018; Rashid et al., 2018). Specifically, many real-world tasks can be divided into interdependent subtasks, with their intricate relationships captured using a directed acyclic graph (DAG) (Shu et al., 2020; Huang et al., 2020; Liu et al., 2023b). Thus a gap exists between methods and applications. This article aims to propose novel algorithms and theories to bridge this gap. More detailed motivation for targeting the DAG setting is provided in Appendix A.

We focus on problems in which subtasks have relationships characterized by a DAG $G := (\mathcal{V}, \mathcal{A})$ where $\mathcal{V}$ and $\mathcal{A}$ denote the set of vertices and the set of arcs, respectively. Arc $(u, v)$ indicates that information flows from $u$ to $v$ such that taking an action for subtask $u$ affects the state of subtask $v$. We formulate our reinforcement learning (RL) problem as a Markov decision process with DAG constraints (MDP-DAG), defined as the tuple $\mathcal{M} = (\{\mathcal{S}^i | i \in \mathcal{V}\}, \{\mathcal{A}^i | i \in \mathcal{V}\}, \{\mathcal{T}^i | i \in \mathcal{V}\}, \{\mathcal{R}^i | i \in \mathcal{L}\}, \{p_0^i | i \in \mathcal{V}\}, \gamma)$, where $\mathcal{L}$ denotes the set of all sinks in the DAG. Each agent $i$ deals with a subtask in the DAG. The transition dynamic $\mathcal{T}^i$ determines the distribution of the next state $s_{t+1}^i$ given the current state $s_t^i$ and the set of actions $\{a_t^j | j \in \Delta(i)\}$, where $\Delta(i)$ is the set of nodes in the sub-graph from the source nodes to node $i$. An agent $i$ for a sink receives a reward $\mathcal{R}^i := r^i(s_t^i, \{a_t^j | j \in \Delta(i))$, where $a_t^j \sim \pi^j(\cdot | s_t^j)$ with $\pi^j$ being the policy for subtask $j$. Let the initial state $s_0^i$ be determined by the distribution $p_0^i$. Then, the objective of learning is to maximize the sum of discounted rewards across all sinks (team rewards), given the structure of the DAG as follows: maximize $\sum_{i \in \mathcal{L}} \mathbb{E}_{\{\pi^j | j \in \Delta(i)\}} \left[ \sum_{t=0}^{\infty} \gamma^t r^i(s_t^i, \{a_t^j | j \in \Delta(i)\}) \right]$, where $\gamma \in [0, 1)$ is the discount factor.

In DAG environments, a high-level agent is highly dependent on the results of lower-level agents (ancestors). As a result, the state and action space of a high-level agent are significantly affected by its ancestors. In particular, in the perspective of a low-level agent, the system does not receive a reward unless all its downstream agents have taken actions. Such a delayed rewarding mechanism is common in many real-world problems including industrial process control (Hein et al., 2018), traffic optimization (Gong et al., 2019), and resource allocation (Xu et al., 2018). Most existing deep reinforcement learning algorithms suffer from inferior performance because no immediate supervision

is given (Gangwani et al., 2019; Liu et al., 2019). Furthermore, a low-level agent cannot directly affect the team's reward, but the team reward depends not only on this agent but also on its descendants. In summary, in DAG environments, it is crucial to consider these complex interrelationships as defined by a DAG.

To address these challenges, we first build a theoretical foundation of our approach. Specifically, we prove that we can at least optimize a lower bound of the optimal value function of the DAG system by introducing the concept of synthetic reward. In addition, to ensure practicality, we propose a new training algorithm that introduces two new entities: leader and reward generator and distributor (RGD) as shown in Fig. 1.

In the proposed approach, the leader generates a goal vector for each follower. The goal is not a human-interpretable goal but an abstract signal that evolves during training so that the leader and the followers utilize it together to communicate for a higher achievement. The leader trains the set of goals for better coordination of followers considering the whole environment, and each follower optimizes its policy by pursuing the given goals.

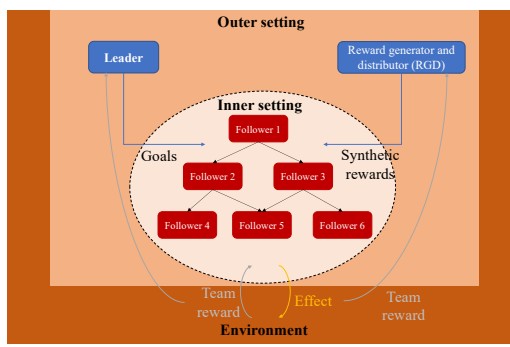

Figure 1: A brief overview of our approach.

In addition, we introduce the concept of the RGD to coordinate agents in the inner setting, called followers, while considering their contributions to the team rewards in the DAG structure. However, the actual contributions of the agents cannot be easily captured through existing non-DAG MARL approaches. In this paper, we develop a strategy to provide incentives (synthetic rewards) using a RGD that generates and distributes reward so that the followers are guided to explore better. Specifically, if a follower contributes to a high team reward, a high synthetic reward is given to the follower by the RGD. Thus, a follower focuses on optimizing its own policy to obtain a high synthetic reward only based on the state of itself. We believe that the concept of the leader and RGD are introduced the very first time herein to address MDP-DAG.

Our main contributions are as follows.

- We propose MARLM-SR to address MDP-DAG by providing a lower bound of the optimal value function based on team rewards under DAG constraints.
- In the proposed learning algorithm, we introduce a novel leader agent to distribute goals to the followers in the form of simple abstract messages that only the leader and the followers can interpret.
- The concept of the reward generator and distributor is first introduced in the area of reinforcement learning to address the problem of reward shaping in the DAG.
- The proposed learning algorithm demonstrates high practicality and scalability because each follower only needs to consider the state of its own subtask.

## 2 RELATED WORKS

**MARL.** In this section, we review MARL studies to address the problem of coordinating non-cooperative agents (referred to as followers). Most existing works focus on simple tabular games or small scale Markov games (Sabbadin & Viet, 2013; 2016; Cheng et al., 2017). Recently, some researchers have proposed deep RL-based leader-follower MARL that can be applied to more general problems. For example, Shu & Tian (2019) applied deep RL to learn an additional agent that assigns sub-tasks to followers with different preferences and skills. However, they limited the environment to cases where the followers are rule-based. Yu et al. (2020) proposed an advanced deep leader-follower MARL algorithm by incorporating a sequential decision module based on the observation that the goal and bonus are sequentially correlated. Jiang and Lu (Jiang & Lu, 2021) introduced a new method termed 'emergence of individuality.' This method employs a probabilistic classifier to predict a probability distribution across multiple agents based on their observations, generating intrinsic reward signals for exploration. Recently, value decomposition schemes, as proposed in (Sunehag et al., 2018; Rashid et al., 2018), have been introduced to assign credits to each agent by decomposing the joint

value function into individual agent-wise value functions. However, these studies do not account for interactions between agents, thereby missing the inherent relationships among subtasks within the context of the entire task. To address this issue, several studies have introduced the concept of a coordination graph, aiming to enhance coordination by capturing locality of interactions (Li et al., 2021; Yang et al., 2022; Kang et al., 2022; Liu et al., 2023a). In these works, the graph represents an implicit coordination relationship among agents for value decomposition based on a specific state, rather than the DAG relationships defined for the entire task. Moreover, these studies assume that agents share the same state and action spaces, making them unsuitable for MDP-DAGs with heterogeneous agents. In summary, to the best of our knowledge, there is no MARL algorithm that can be used to coordinate multiple agents in a DAG defined within the context of the entire task, which is our target.

**Reward shaping for multi-agent systems.** Often, environmental feedback is not enough to effectively train an agent, especially when the environment is stochastic (Devlin & Kudenko, 2016). In this case, reward shaping can help guide an agent's exploration by providing an additional artificial reward signal. A few researchers have proposed reward shaping methods for multi-agent systems. Colby et al. (2015) showed that their algorithm called 'difference rewards' is powerful in effectively allocating rewards across multiple agents. Here, 'difference rewards' was designed to reveal the contribution of the current action of an agent by comparing the current reward to the reward received when the action of the agent is replaced with the default action (Wolpert & Tumer, 2001). In practice, 'difference rewards' can be estimated using a function approximation technique (Foerster et al., 2018). It has been proven that potential-based reward shaping, which is one of the typical reward shaping methods, does not alter the optimal policy (Ng et al., 1999). Based on this background, Devlin et al. (2014) proposed two potential-based reward shaping methods based on 'difference rewards.' Even though this algorithm guarantees optimality, it assumes top-down MARL, in which all agents have a common task and a centralized system distributes rewards to the agents based on their contributions. Thus, it lacks scalability and applicability. To tackle this problem, Aotani et al. (2021) proposed a localized reward shaping method that prevents the agents from knowing the interests between them. However, this work still cannot consider the relationship between agents in a DAG.

## 3 MODELING SETTING

Global decision-making is mainly used for many real-world systems. However, traditional global single-agent RL models (GSARLMs) are poorly suited to environments under DAG constraints even though the global model can provide an optimal or a very good solution theoretically (Lowe et al., 2017). This is because, in general, the search space for obtaining a single global solution is too large while compromising scalability. In addition, GSARLM cannot easily capture interactions between multiple subtasks in a DAG. Thus, in this section, we define the MARL model with synthetic rewards (MARLM-SR) and build an analytical background. In addition, we further decompose the problem by introducing the concept of goal periods. Finally, we provide strong evidence of higher practicality and scalability of MARLM-SR based on this decomposed problem by proposing a training algorithm in the next section. Given the introduction of numerous new terms within our modeling setting, we provide an illustrative example in Appendix B to enhance comprehension.

### 3.1 MARLM-SR

The objective of GSARLM is to derive an optimal solution that covers all subtasks considering the current states of all subtasks altogether. Even though one action is made to cover all subtasks, the state transition of each subtask is stochastically determined based on inherent DAG relationships. Let $s_t^i$ and $a_t^i$ be the state and action of subtask $i \in \mathcal{V}$. First, the lowest-level subtasks, the source nodes $i$ in the DAG, are affected only by themselves based on the stochastic state transition $s_{t+1}^i \sim p(\cdot|s_t^i, a_t^i)$. On the other hand, the states of the other subtasks are affected by the ancestor nodes in the DAG, $s_{t+1}^i \sim p(\cdot|s_t^i, \{a_t^j | j \in \Delta(i)\})$.

Let us assume that $\Pi$, the policy for the entire system, can be decomposed into $(\pi^1, \pi^2, \cdots, \pi^I)$ in which $\pi^i$ is the policy for subtask $i$, where $I = |\mathcal{V}|$. In addition, since the performance of a system with a DAG structure is represented by the rewards of the sinks in the DAG, the highest-level subtasks, we assume that the team reward is the sum of the rewards obtained from sinks. Let $r^i$ be the reward function of subtask $i$, $i \in \mathcal{L}$, the set of all sinks. Then, we define the value function of a subtask $i$ in $\mathcal{L}$

as follows

$$V_i^{\{\pi^j|j\in\Delta(i)\}}(s_0^i) = \mathbb{E}_{\{\pi^j|j\in\Delta(i)\}}\left[\sum_{t=0}^{\infty}\gamma^t r^i(s_t^i, \{a_t^j|j\in\Delta(i)\})\right] \tag{1}$$

where $V_i^{\{\pi^j|j\in\Delta(i)\}}$, the value of subtask $i$, has dependency on $\{\pi^j|j\in\Delta(i)\}$. The objective function is $\underset{\pi^1,\cdots,\pi^I}{\text{maximize}} \sum_{i\in\mathcal{L}} V_i^{\{\pi^j|j\in\Delta(i)\}}(s_0^i)$.

Next, we introduce the concept of MARL with synthetic rewards. First, an agent deals with its own subtask and receives a synthetic reward. Here, we assume that the synthetic reward for an agent is determined by considering its contribution to the team rewards. In other words, an agent's policy which contributes to a high reward of its descendant sinks yields a high synthetic reward. We assume that there can be a function $f_{ik}$ that measures the contribution of agent $i$ to sink agent $k$'s reward and the total contribution of agents in $\Delta(k)$ to sink agent $k$'s reward is less than or equal to 1 as shown in (2) because the reward of a sink agent is also affected by environmental feedbacks. All subtasks that have a path to/from subtask $i$ have an impact on the agent $i$'s contribution. Thus, the synthetic reward function of agent $i$ has dependency on $\Omega(i) = \Delta(i) \cup \Upsilon(i)$, where $\Upsilon(i)$ denotes the set of subtasks in the induced sub-graph rooted in subtask $i$ including node $i$. Finally, we have the following definition.

**Definition 1** Let $f_{ik}$ be a function that produces the magnitude of agent $i$'s contribution to sink agent $k$'s reward for $k \in \Upsilon(i)$. For any $f_{ik}$ satisfying

$$\sum_{i\in\Delta(k)} f_{ik}((s_t^j, a_t^j)|j\in\Delta(k)) \le 1 \ \forall k\in\mathcal{L}, \tag{2}$$

the synthetic reward function $sr^i$ of subtask $i$ is defined as

$$sr^i((s_t^j, a_t^j)|j\in\Omega(i)) = \sum_{k\in\mathcal{L}\cap\Upsilon(i)} f_{ik}((s_t^j, a_t^j)|j\in\Delta(k))r^k(s_t^k, \{a_t^j|j\in\Delta(k)\}) \ \forall i\in\mathcal{V}. \tag{3}$$

**Definition 2** We define synthetic value functions based on synthetic rewards as

$$\tilde{V}_i^{\{\pi^j|j\in\Omega(i)\}}(s_0^i) = \mathbb{E}_{\{\pi^j|j\in\Omega(i)\}}\left[\sum_{t=0}^{\infty}\gamma^t sr^i((s_t^j, a_t^j)|j\in\Omega(i))\right] \ \forall i\in\mathcal{V}. \tag{4}$$

Next, we show that the total synthetic value provides a lower bound on the total value; thus, we can optimize agents' policies such that synthetic values are maximized in order to maximize a lower bound of the sum of optimal values. It provides the theoretical background that we only need to train agents to seek high synthetic rewards in a parallel fashion. In Section 4, we propose a practical algorithm for generating and distributing synthetic rewards.

**Theorem 1** If reward $r^i \ge 0$, $\forall i\in\mathcal{L}$, then, for any $f_{ik}$ satisfying (2), we have

$$\sum_{i\in\mathcal{V}} \tilde{V}_i^{\{\pi^j|j\in\Omega(i)\}}(s_0^i) \le \sum_{i\in\mathcal{L}} V_i^{\{\pi^j|j\in\Delta(i)\}}(s_0^i). \tag{5}$$

*Proof.* A detailed proof of this theorem is given in Appendix C.

## 3.2 MARLM-SR with goal period

We further extend MARLM-SR by introducing the notion of a goal period, which is a short interval that partitions an episode, enabling more refined coordination between agents over the learning process using two novel entities: leader and RGD. Let $D$ be the number of steps for a goal period, and $s_{ld}^i$ and $a_{ld}^i$ be the state and action at $d$-th step in $l$-th goal period, respectively. As a consequence (1) and (4) change to $V_i^{\{\pi^j|j\in\Delta(i)\}}(s_{01}^i) = \mathbb{E}_{\{\pi^j|j\in\Delta(i)\}}\left[\sum_{l=0}^{\infty}\sum_{d=1}^{D}\gamma^{lD+d-1} r^i(s_{ld}^i, \{a_{ld}^j|j\in\Delta(i)\})\right] \ \forall i\in\mathcal{L}$ and

$\tilde{V}_i^{\{\pi^j|j\in\Omega(i)\}}(s_{01}^i) = \mathbb{E}_{\{\pi^j|j\in\Omega(i)\}}\left[\sum_{l=0}^{\infty}\sum_{d=1}^{D}\gamma^{lD+d-1} sr^i((s_{ld}^j, a_{ld}^j)|j\in\Omega(i))\right] \ \forall i\in\mathcal{V}$, respectively.

From these two equations and Theorem 1, we obtain

$$\max_{\{f_{ik}|k\in\mathcal{L}, i\in\Delta(k)\}} \sum_{i\in\mathcal{V}} \tilde{V}_i^{\{\pi^j|j\in\Omega(i)\}}(s_{01}^i) \le \sum_{i\in\mathcal{L}} V_i^{\{\pi^j|j\in\Delta(i)\}}(s_{01}^i), \tag{6}$$

subject to $\{f_{ik}|k\in\mathcal{L}, i\in\Delta(k)\}$ complying to Definition 1. This is the basis of our algorithm presented in the next section.

# 4 ALGORITHM

In this section, we describe the training algorithm for MARLM-SR. The algorithm consists of the outer and inner settings. In the inner setting, the followers perform their subtasks given by the defined DAG every time step. On the other hand, in the outer setting, two different types of agents are trained to guide the followers to achieve a high team reward. If the followers are guided well based on the policies of the outer agents and a high team reward is achieved, this high team reward is given to the outer agents. We provide a more detailed exposition of the algorithm, including its pseudo-code, in Appendix D.

## 4.1 OUTER SETTING

The leader provides a different goal to each follower at the beginning of each goal period. It is governed by an RL model with policy $\pi^L$. Here, the goal is a vector with fixed length in which each element has a value between 0 and 1. It is used for communication between the leader and the followers. Since the leader is rewarded based on the followers' achievements, it must be trained to produce meaningful goals. On the other hand, the followers must interpret the goals and use this information to achieve high team rewards. Let $S_{ld} = (s_{ld}^i | i \in \mathcal{V})$ be the global state at step $d$ and $G_l = (g_l^i | i \in \mathcal{V})$ be the set of goals in the $l$-th goal period. Each follower augments its state with $g_l^i$ and thus the state of follower $i$ at step $d$ is $\overline{s}_{ld}^i = (s_{ld}^i, g_l^i)$. In addition, the RGD is modeled with policy $\pi^{RGD}$ that produces synthetic reward $sr_l^i$ for each follower $i$ after the $l$-th goal period (details for generating $sr_l^i$ are provided later in this section).

The leader is trained to produce $G_l$ that maximizes team rewards since the team rewards are also given to the leader as its own reward. The leader receives cumulative team rewards after each goal period. Thus, the reward of the leader after the $l$-th goal period is defined as $\sum_{i \in \mathcal{L}} \sum_{d=1}^{D} r^i(s_{ld}^i, \{a_{ld}^j | j \in \Delta(i)\})$. By extending this cumulative reward to cover infinite goal periods, the objective function for the leader is defined as

$$\underset{\pi^L}{\text{maximize }} V_L^{\{\pi^L, \pi^{RGD}, \pi^j | j \in \Delta(i)\}}(S_{01}) = \sum_{i \in \mathcal{L}} \mathbb{E}_{\{\pi^L, \pi^{RGD}, \pi^j | j \in \Delta(i)\}} \left[ \sum_{l=0}^{\infty} \gamma^l \sum_{d=1}^{D} r^i(s_{ld}^i, \{a_{ld}^j | j \in \Delta(i)\}) \right],$$
(7)

where the state transition of $S_{ld}$ (in a particular goal period) depends on the underlying policies. The state of the leader is defined as $S_l^L = S_{l1} \circ (g_{l-1}^i | i \in \mathcal{V}) \circ (sr_{l-1}^i | i \in \mathcal{V})$, including the initial global state in each goal period $l$. By $\circ$ we denote the concatenation operator. Then, the state transition of the leader is defined as $S_{l+1}^L \sim p(\cdot | S_l^L, \{a_{ld}^i | i \in \mathcal{V} \text{ and } d = 1, \cdots, D\}, (g_l^i | i \in \mathcal{V}), (sr_l^i | i \in \mathcal{V})) \circ (g_l^i | i \in \mathcal{V}) \circ (sr_l^i | i \in \mathcal{V})$. Additionally, the set of goals are produced based on $(g_l^i | i \in \mathcal{V}) \sim \pi^L(\cdot | S_l^L)$.

The RGD should be able to figure out the followers' state changes to provide effective coordination strategies. The easiest way is to collect the global state for all time steps in a goal period and use it as the input state. However, to prevent the RGD's input from being too high dimensional, we sample global states with equal time step intervals including the first and last global states in a goal period. For simplicity, we call the set of sampled global states as the global state flow (GSF). This state GSF is defined as $gsf_l = (S_{l,kj+1} | j = 0, \cdots, \lfloor \frac{D-1}{k} \rfloor) \circ S_{l+1,1}$, where $k$ is a hyperparameter and $\lfloor \cdot \rfloor$ is the floor function. Vector $S_{l+1,1}$ is the global state after the last action set $\{a_{l,D}^i | i \in \mathcal{V}\}$ is taken in the $l$-th goal period. Goals are also used to guide the RGD; thus, the state of the RGD is $S_l^{RGD} = gsf_l \circ (g_l^i | i \in \mathcal{V})$. The state transition of the RGD is defined as $S_{l+1}^{RGD} \sim p(\cdot | gsf_l, \{a_{l+1,d}^i | i \in \mathcal{V} \text{ and } d = 1, \cdots, D\}, (g_{l+1}^i | i \in \mathcal{V}), (sr_l^i | i \in \mathcal{V})) \circ (g_{l+1}^i | i \in \mathcal{V})$.

The RGD policy produces a team reward signal $q_l$, node values $(v_l^i | i \in \mathcal{V})$, and arc values $(e_l^{(i,j)} | (i,j) \in \mathcal{A})$ for synthetic reward generation and distribution. All these values are within the range [0, 1]. The policy is specified by $(q_l) \circ (v_l^i | i \in \mathcal{V}) \circ (e_l^{(i,j)} | (i,j) \in \mathcal{A}) \sim \pi^{RGD}(\cdot | S_l^{RGD})$. Vector $(sr_l^i | \forall i \in \mathcal{V})$ is obtained based on $(q_l)$, $(v_l^i | i \in \mathcal{V})$, and $(e_l^{(i,j)} | (i,j) \in \mathcal{A})$, not by a closed-form function, but by the proposed reward generation and distribution algorithm exhibited next.

The synthetic reward $sr_l^i$, $i \in \mathcal{V}$, is given to the followers as a bonus after each goal period. The RGD should provide a high synthetic reward if followers use policies that lead to high team rewards. In addition, the value of the synthetic reward must be adjusted dynamically to make the policy of

the RGD significant. This is because followers are more likely to achieve higher team rewards as training progresses. In this case, the same reward can be too small for followers who have had enough training but can be too large for followers without enough training. The quality of the learned policy is revealed as the team reward of the previous episode. The RGD policy produces $q_l$ (in addition to $v$ and $e$). This value is multiplied with $\frac{\overline{R}_e}{\overline{N}_e}$, the average team reward per goal period, in the previous episodes, where $\overline{N}_e$ is the average number of goal periods and $\overline{R}_e$ is the average total team reward. Finally, in the current episode, the total synthetic reward after the $l$-th goal period is $M_l = q_l \frac{\overline{R}_e}{\overline{N}_e}$. We simply set $\overline{R}_0 = 0$ or a negligible value.

We assume that the synthetic reward for the follower $i$ is determined based on its contributions to the sink followers among its descendants and their rewards as defined in (3). Thus, we propose a synthetic reward distribution strategy that first sets synthetic reward portions for the followers in sinks considering their achievements, and then sends them down to account for the contributions of lower-level followers. The RGD is trained to achieve high team rewards by creating a good distribution strategy because it is quite challenging to estimate the exact contribution of each agent.

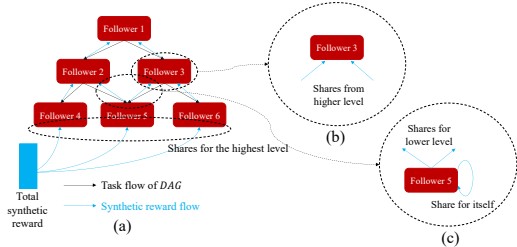

Figure 2: An overview of reward distribution. The algorithm first determines the shares for the highest-level followers. Then, one level lower followers receive the shares set as shown in (b). After receiving the rewards in (c), the shares for themselves and for the one level lower followers are determined. This process is repeated until the lowest-level followers receive their shares.

The RGD distributes the synthetic reward generated by the reward generator as shown in Fig. 2. Because the synthetic reward flows in the opposite direction of the task flow, arc $(i, j)$ denotes a directed edge from a higher-level node $i$ to a lower-level node $j$. We can sequentially calculate shares from the highest-level to the lowest-level followers. First, we calculate initial share $\tilde{sh}_l^i$ for a highest-level follower $i \in \mathcal{L}$ after goal period $l$ as $\tilde{sh}_l^i = \begin{cases} \frac{v_l^i}{\sum_{k \in \mathcal{L}} v_l^k}, & \text{if } \sum_{k \in \mathcal{L}} v_l^k > 0 \\ \frac{1}{|\mathcal{L}|}, & \text{otherwise} \end{cases}$, where $|\mathcal{L}|$ is the number of the sinks. Similarly, for each follower, the initial share can be determined after receiving all the rewards from one level higher followers. After all children of the agent $i$ determine the share to the agent $i$, the initial share $\tilde{sh}_l^i$ is simply calculated by $\tilde{sh}_l^i = \sum_{k \in ch(i)} sh_l^{k,i}$, where $sh_l^{k,i}$ is the share from $k$ to $i$. After $\tilde{sh}_l^i$ is determined, the final reward shares to the follower $i$ itself and the arc $(i, j)$ are defined as $sh_l^i = \begin{cases} \tilde{sh}_l^i \times \frac{v_l^i}{v_l^i + \sum_{j \in \delta(i)} e_l^{i,j}}, & \text{if } v_l^i + \sum_{j \in \delta(i)} e_l^{i,j} > 0 \\ \frac{1}{1+|\delta(i)|}, & \text{otherwise} \end{cases}$ and

$sh_l^{i,j} = \begin{cases} \tilde{sh}_l^i \times \frac{e_l^{i,j}}{v_l^i + \sum_{j \in \delta(i)} e_l^{i,j}}, & \text{if } v_l^i + \sum_{j \in \delta(i)} e_l^{i,j} > 0 \\ \frac{1}{1+|\delta(i)|}, & \text{otherwise} \end{cases}$, respectively. Here, $\delta(i)$ denotes the parents of the follower $i$. After $sh_l^i$ is determined for all $i \in \mathcal{V}$, $sr_l^i = sh_l^i M_l$ is provided to agent $i$ as the synthetic reward after the goal period $l$.

Same as the leader, the RGD is trained with the aim of maximizing team rewards by obtaining better coordination through synthetic rewards. However, since the first action of the RGD is taken after the first goal period, we define the value function for the RGD as (8) and train the RGD to maximize it.

$$V_{RGD}^{\{\pi^L, \pi^{RGD}, \pi^j | j \in \Delta(i)\}}(gsf_0) = \sum_{i \in \mathcal{L}} \mathbb{E}_{\{\pi^L, \pi^{RGD}, \pi^j | j \in \Delta(i)\}} \left[ \sum_{l=1}^{\infty} \gamma^{l-1} \sum_{d=1}^{D} r^i(s_{ld}^i, \{a_{ld}^j | j \in \Delta(i)\}) \right], \quad (8)$$

## 4.2 INNER SETTING

In the inner setting, the followers are trained with the supervision of the outer agents. Because the goal given by the leader is incorporated into the state, state transition is defined as $\overline{s}_{l,d+1}^i \sim p(\cdot | \overline{s}_{ld}^i, \{a_{ld}^j | j \in$

$\Delta(i)\}$). In each episode during training, followers' achievements are rewarded in two ways. First, the followers share the team reward equally because it is not only quite challenging to create synthetic rewards based on the exact contribution to the team reward, but the team reward can also serve as effective supervision. For each follower, $\frac{\sum_{i \in \mathcal{L}} r^i(s^i_{ld}, \{a^j_{ld} | j \in \Delta(i)\})}{|\mathcal{V}|}$ is given as a shared team reward at the $d$-step of the $l$-th goal period. In addition, the follower $i$ receives a synthetic reward $sr^i_l$ from the RGD after the $l$-th goal period based on the difference in their achievements. By considering both the shared team reward and the synthetic reward, we define the objective function of the follower $i$ as

$$\underset{\pi^i}{\text{maximize}} \; \overline{V}_i^{\{\pi^L, \pi^{RGD}, \pi^j | j \in \mathcal{V}\}}(\overline{s}^i_{01}) =$$

$$\mathbb{E}_{\{\pi^L, \pi^{RGD}, \pi^j | j \in \mathcal{V}\}} \left[ \sum_{l=0}^{\infty} \left[ \gamma^{(l+1)D-1} sr^i_l + \sum_{d=1}^{D} \sum_{k \in \mathcal{L}} \gamma^{lD+d-1} \frac{r^k(s^k_{ld}, \{a^u_{ld} | u \in \Delta(k)\})}{|\mathcal{V}|} \right] \right]. \quad (9)$$

Here, we use $\overline{V}$ to distinguish it from the value functions in the modeling section, which only consider explicit rewards or synthetic rewards.

In the algorithm, the leader sets goals at the beginning of each goal period and is rewarded after the goal period. On the other hand, the RGD determines the synthetic reward distribution strategy after each goal period. And this strategy influences the followers to behave differently in the next goal periods. Therefore, the RGD is rewarded in the next goal period.

## 5 Experiments

**Implementation details.** We used a proximal policy optimization algorithm (Schulman et al., 2017b) to optimize the policies of all agents used in this work. Additional implementation details, including the hyperparameters used for the proposed algorithm and the baselines, are summarized in Appendix F. We have open sourced the code at `https://github.com/n2kdnk1123/MARLM-SR`.

**Environments.** We created three artificial environments to simulate systems with DAG constraints: a factory production planning case, a logistics case, and a hierarchical predator-prey case. We also investigated the performance of the proposed algorithm in real-world scheduling for one of Intel's high volume packaging and test factories. The details of all environments are described in Appendix E. For confidentiality reasons, we offer public access to only the three artificial environments.

### 5.1 Baselines

We have compared seven baseline algorithms against our algorithm. First, we used the following five algorithms that do not employ reward shaping.

- Global single-agent algorithm (GS): In this baseline, a single agent is learned to do all subtasks.
- Shared reward multi-agent algorithm (SRM): Each agent deals with a subtask and shares the reward. This algorithm is perhaps the most popular multi-agent learning algorithm, also known as independent Q-learning (Tan, 1993) or independent actor-critic (Foerster et al., 2018), depending on the type of the learner used.
- Leader-follower multi-agent algorithm (LFM): This baseline adds the leader to SRM. Specifically, the followers are given the goals as well as the shared rewards.
- RGD-follower multi-agent algorithm (RFM): The RGD is added to SRM in this baseline. Thus, the followers are given the synthetic rewards as well as the shared team rewards.
- The proposed algorithm: We have the leader, the RGD, and the followers of the proposed algorithm. This algorithm adds the RGD to LFM and the leader to RFM.

The last two are our stripped-down algorithms and, as such, not previously existing algorithms. We are not aware of any reward shaping method targeting DAGs, however we found two existing reward shaping methods that can be applied to coordinate multiple agents. We introduced these two reward shaping methods to the MARL algorithm that trains agents in parallel. Specifically, we also compared the following two baselines against ours.

- Difference rewarding method (Colby et al., 2015) + MARL algorhitm (Diff-M)
- Counterfactual as Potential (Devlin et al., 2014) + MARL algorhitm (CaP-M)

## 5.2 RESULTS

In our proposed algorithm, the outer agents are trained to coordinate followers by providing additional synthetic rewards that correspond to the contributions of the followers in the DAG. To ascertain the effectiveness of reward shaping, we initially evaluated the proposed algorithm against Diff-M and CaP-M. Fig. 3 shows comparison results on the three artificial benchmark cases. The plots use the moving window method, which averages the team rewards over 100 episodes with a step size of one, to reduce variability. The standard deviation is represented as a shaded area. The results demonstrate that our method achieves significantly superior performance across all three benchmark cases. Specifically, in terms of the average team reward over the last 100 episodes for the three artificial cases, the proposed algorithm achieves performance that is 132.7% and 89.3% higher than that of Diff-M and CaP-M, respectively. This suggests that, until now, there has not been an effective reward shaping method for systems under DAG constraints. In the case of logistics, ours quickly get away from a bad local optima where agents send almost nothing to the next level agents to reduce inventory cost (refer to Appendix E), even after it get stuck in.

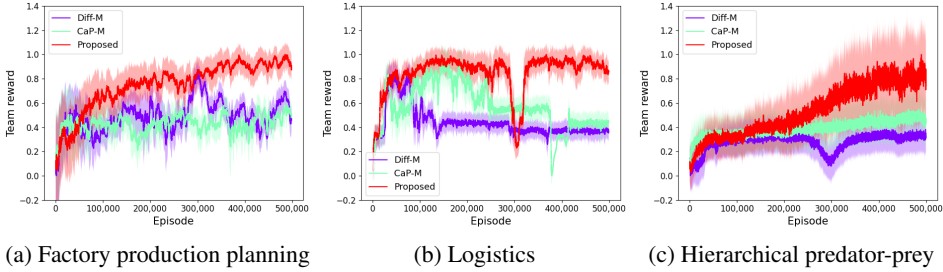

(a) Factory production planning       (b) Logistics       (c) Hierarchical predator-prey

Figure 3: Comparison with state-of-the-art algorithms on the three artificial benchmark cases. Min-max normalization is applied to the team reward to standardize the scale of the y-axis.

We also compared the two baseline algorithms across diverse scheduling scenarios. Specifically, we trained the agents in the DAG using the proposed algorithm, Diff-M, and CaP-M and then evaluated their performance on 1,000 new scheduling scenarios (episodes). Fig. 4 presents the histogram comparing the completion rates of the three baselines. In the histograms, we omitted the labeling of x-axis values for confidentiality reasons; however, all histograms share the same scale, with equally spaced intervals along the x-axis. From the results it is clear that our proposed algorithm achieves higher overall completion rates. Specifically, the proposed algorithm demonstrated a performance improvement of 19.2% and 4.4% in terms of the mean completion rate, compared to Diff-M and CaP-M, respectively. In summary, our proposed method of synthetic reward generation and distribution, coupled with communication through the leader's goals, can enhance coordination leading to increased team rewards.

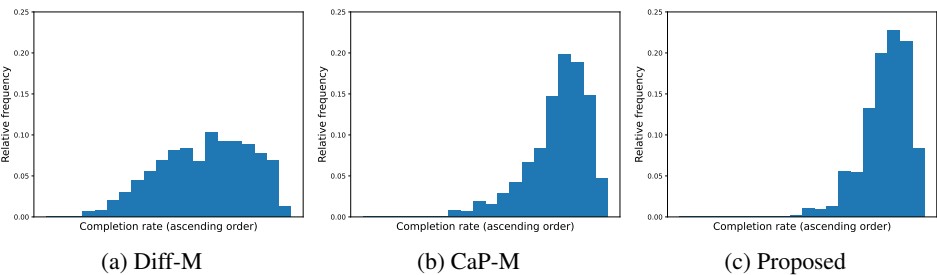

(a) Diff-M       (b) CaP-M       (c) Proposed

Figure 4: The histogram of the completion rate over 1,000 scheduling scenarios (episodes) for comparison with the state-of-the-art algorithms.

We also conducted ablation studies to evaluate the effectiveness of each component in the proposed algorithm. Fig. 5 shows the comparison results of the five baselines: GS, SRM, LFM, RFM, and our proposed algorithm, on the three artificial benchmark cases. Specifically, GS shows the worst performance in all three cases, revealing that introducing the multi-agent concept is effective for environments with DAG constraints. The leader can help improve performance as shown in (b) and (c). However, by comparing LFM and RFM, we find that the RGD contributes more to performance improvement than the leader in (a) and (c). Specifically, on average over the three cases, LFM and

RFM improve the average team reward over the last 100 episodes by 5.6% and 56.0% compared to SRM, respectively. Nonetheless, the proposed algorithm demonstrates the best learning curve in all settings, while achieving an 82.4% higher average team reward compared to SRM. In addition, the performances of LFM and RFM in Fig. 5 are overall better than those of Diff-M and CaP-M in Fig. 3. In other words, we are able to achieve better performance only by adding one component, either the leader or the RGD, in DAG environments. In addition, combining the two components further enhances performance.

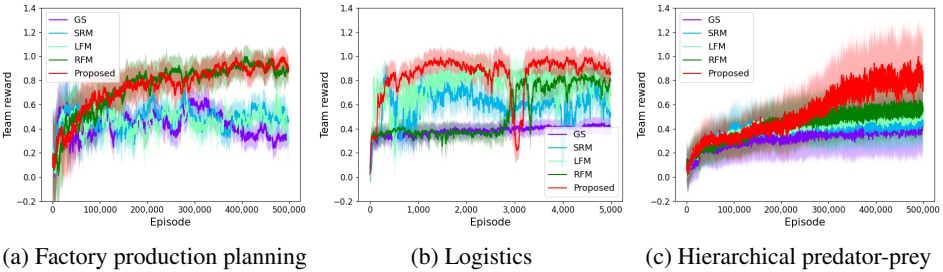

(a) Factory production planning      (b) Logistics      (c) Hierarchical predator-prey

Figure 5: Learning curves of five baselines for ablation study. Min-max normalization is applied to the team reward to standardize the scale of the y-axis.

The five baselines are also compared in diverse scheduling scenarios. The histogram of the completion rate for the five baselines, along with the results of statistical significance tests, can be found in Appendix G. The result demonstrates the significant superiority of the proposed algorithm over the other baselines except for LFM. The proposed algorithm achieves a performance improvement of 3.9% by introducing the RGD, and an improvement of 8.5% by introducing both the leader and the RGD together. Even though LFM achieved a good performance similar to ours, the contribution of the RGD is not negligible considering the results in Fig. 5. Thus, we can state both the leader and the RGD are necessary for our algorithm. A more detailed discussion is provided in Appendix G.

Finally, we conducted sensitivity analyses on the length of the goal period using the three artificial benchmark cases. We established four length levels: short, medium, long, and extremely long for each case. The details of the sensitivity analysis settings and results, including the specified length for each level, are provided in Appendix H. Fig. 6 illustrates that the goal period length should not be excessively long, as it can result in poor coordination among followers by the outer agents. However, a short goal period does not always guarantee optimal performance, so the length should be adjusted based on the specific environment. We also conducted an analysis of sensitivity concerning the dimension of the goal vector; the detailed results can be found in Appendix H. In summary, the results suggest that significant performance gains are attainable when the goal vector has a limited dimension, but the gains rapidly decrease as the dimension increases.

## 6 DISCUSSION

In this paper, a theoretical background on MARLM-SR was established and a novel training algorithm for coordinating multiple agents in a DAG environment was proposed. Comparison results in several DAG environments including a real-world scheduling environment confirmed that our approach significantly outperforms existing non-DAG algorithms. It was found that the leader and the RGD contributed to this overwhelming performance. One limitation of this work is that we did not provide a mathematical basis for whether the synthetic reward obtained through our algorithm satisfies the conditions in the modeling section. Instead, the superiority of the proposed algorithm was shown through empirical results. Nonetheless, there have been few opportunities to apply our algorithm to real-world industrial cases. Therefore, in future studies, the proposed algorithm will be further developed by applying it to more diverse real-world industrial cases.

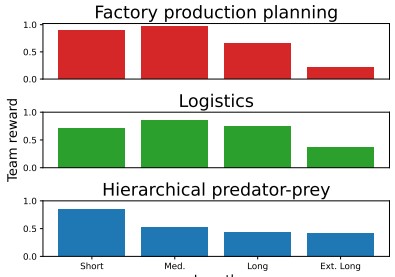

Figure 6: Sensitivity analysis results. We use the average team reward over the last 10,000 episodes during training. Specified length for each level can be found in Appendix H.

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

## A    MOTIVATION

In our study, we applied the proposed algorithm to a real-world scheduling scenario in Intel's high-volume packaging and test factory after modeling the factory as a DAG based on the precedence constraints of subtasks. Scheduling is crucial in many semiconductor manufacturing processes (and other high-tech industries) to maintain high productivity. Considering the semiconductor industry's market size (forecasted to be $625 billion in 2023 and $1,307 billion in 2032 (market.us, 2023)), it is important to address the scheduling challenges using a DAG-based approach. Furthermore, many IoT applications in cyber-physical systems, such as smart grids, require control actions for multiple agents following a DAG structure (Khare et al., 2019). With industry systems becoming increasingly complex, the DAG setting is likely to become more prevalent in real-world scenarios.

Fig. 7 illustrates the DAG of a simplified car manufacturing process. In this process, there are several subtasks in the production line. Each subtask is managed by a unique agent. Initially, agent 1 collects raw materials such as metal sheets, rubber, plastic, and more. Once these raw materials are collected, agent 2 begins crafting the metal framework. Following the readiness of these agents, agents 3 and 4 undertake engine assembly and exterior work respectively. After the exterior is finished and the engine is assembled, agent 5 commences the interior setup. Once the interior is complete, the car is sent to agent 6 for painting and finishing touches. In this example, each agent cannot commence until lower-level agents have taken appropriate actions to send the completed parts. Additionally, regardless of the number of parts completed or the quality achieved by low-level agents, a high team reward cannot be attained unless the higher-level agents successfully perform their subtasks. In other words, all agents must be coordinated towards the team goal, taking into account the interrelationships defined by the DAG.

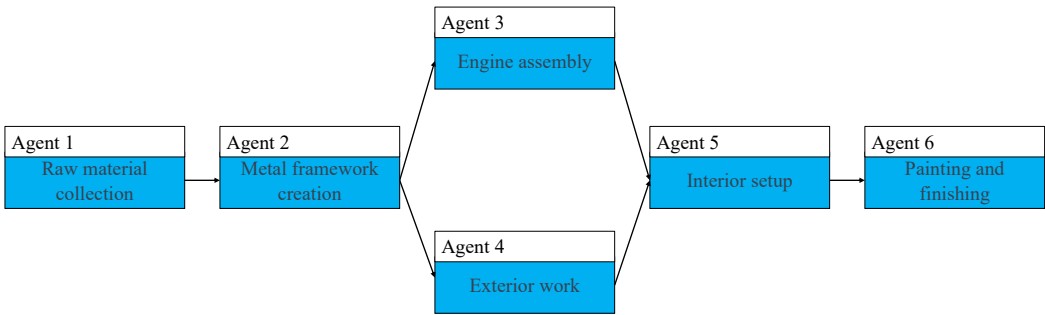

Figure 7: The DAG of a simplified car manufacturing process

## B    ILLUSTRATIVE EXAMPLE OF THE MODELING SETTING

In this section, we present an illustrative example of the modeling setting to enhance comprehension. For this illustration, we refer to the case shown in Fig. 7. To simplify the exposition, we do not incorporate the concept of goal periods in this example. In this scenario, Agent 1 serves as the sole source node, and thus, the state transition probability for this agent is defined as $s_{t+1}^1 \sim p(\cdot|s_t^1, a_t^1)$. However, the state transition probabilities for the other agents are defined with consideration of the agents in the ancestor nodes. For instance, the state transition probability of Agent 5 is defined as $s_{t+1}^5 \sim p(\cdot|s_t^5, \{a_t^j|j \in \{1, 2, 3, 4, 5\}\})$, taking into account ancestors from Agent 1 to Agent 4.

Next, we have a single sink node, Agent 6, which has all other nodes as ancestors in this example. Therefore, we can define the value function in (1) for Agent 6 as

$$V_6^{\{\pi^j|j\in\{1,2,3,4,5,6\}\}}(s_0^6) = \mathbb{E}_{\{\pi^j|j\in\{1,2,3,4,5,6\}\}}\left[\sum_{t=0}^{\infty}\gamma^t r^6(s_t^6, \{a_t^j|j\in\{1,2,3,4,5,6\}\})\right]. \quad (10)$$

The objective function for this example is to maximize $V_6^{\{\pi^j|j\in\{1,2,3,4,5,6\}\}}(s_0^6)$ with respect to $\{\pi^1, \cdots, \pi^6\}$.

We have also introduced the concept of synthetic reward and value functions for not only the agents in sink nodes but also for all agents. To compute these two functions, we consider the dependency of agent $i$ on $\Omega(i) = \Delta(i) \cup \Upsilon(i)$, where $\Delta(i)$ and $\Upsilon(i)$ represent the set of nodes in the sub-graph from the source nodes to node $i$ and the set of subtasks in the induced sub-graph rooted at subtask $i$, including node $i$, respectively.

Let us consider Agent 4 as an example, and let $f_{ik}$ be a function that calculates the magnitude of agent $i$'s contribution to the reward of sink agent $k$ for $k \in \Upsilon(i)$. In this case, $\Delta(4) = \{1, 2, 4\}$, $\Delta(6) = \{1, 2, 3, 4, 5, 6\}$, and $\Upsilon(4) = \{4, 5, 6\}$. Consequently, the synthetic reward function in (3) is defined as follows :

$$sr^4((s_t^j, a_t^j)|j \in \{1,2,4,5,6\}) = f_{4,6}((s_t^j, a_t^j)|j \in \{1,2,3,4,5,6\})r^6(s_t^6, \{a_t^j|j \in \{1,2,3,4,5,6\}\}). \quad (11)$$

Here, we assume that the state and action of Agent 3, which are not included in $\Omega(4)$, are given. Finally, the synthetic value function in (4) is defined as

$$\tilde{V}_4^{\{\pi^j|j\in\{1,2,4,5,6\}\}}(s_0^4) = \mathbb{E}_{\{\pi^j|j\in\{1,2,4,5,6\}\}}\left[\sum_{t=0}^{\infty}\gamma^t sr^4((s_t^j, a_t^j)|j \in \{1,2,4,5,6\})\right]. \quad (12)$$

## C    PROOF OF THEOREM 1

For simplicity, let $g_{ik} = \sum_{t=0}^{\infty}\gamma^t f_{ik}((s_t^j, a_t^j)|j \in \Delta(k))r^k(s_t^k, \{a_t^j|j \in \Delta(k)\})$. We have

$$\sum_{i\in\mathcal{V}}\tilde{V}_i^{\{\pi^j|j\in\Omega(i)\}}(s_0^i)$$
$$= \sum_{i\in\mathcal{V}}\mathbb{E}_{\{\pi^j|j\in\Omega(i)\}}\left[\sum_{t=0}^{\infty}\gamma^t sr^i((s_t^j, a_t^j)|j \in \Omega(i))\right]$$
$$= \sum_{i\in\mathcal{V}}\mathbb{E}_{\{\pi^j|j\in\Omega(i)\}}\left[\sum_{t=0}^{\infty}\gamma^t\sum_{k\in\mathcal{L}\cap\Upsilon(i)}f_{ik}((s_t^j, a_t^j)|j \in \Delta(k))r^k(s_t^k, \{a_t^j|j \in \Delta(k)\})\right]$$
$$= \sum_{i\in\mathcal{V}}\mathbb{E}_{\{\pi^j|j\in\Omega(i)\}}\left[\sum_{k\in\mathcal{L}\cap\Upsilon(i)}g_{ik}\right]. \quad (13)$$

Since $g_{ik}$ has dependency only on $\{\pi^j|j \in \Delta(k)\}$ and $f_{ik}((s_t^j, a_t^j)|j \in \Delta(k))$, we further have

$$\sum_{i\in\mathcal{V}}\mathbb{E}_{\{\pi^j|j\in\Omega(i)\}}\left[\sum_{k\in\mathcal{L}\cap\Upsilon(i)}g_{ik}\right]$$

$$=\sum_{i\in\mathcal{V}}\sum_{k\in\mathcal{L}\cap\Upsilon(i)}\mathbb{E}_{\{\pi^j|j\in\Delta(k)\}}\left[g_{ik}\right]$$

$$=\sum_{k\in\mathcal{L}}\mathbb{E}_{\{\pi^j|j\in\Delta(k)\}}\left[\sum_{i\in\Delta(k)}g_{ik}\right]$$

$$=\sum_{k\in\mathcal{L}}\mathbb{E}_{\{\pi^j|j\in\Delta(k)\}}\left[\sum_{t=0}^{\infty}\gamma^t\sum_{i\in\Delta(k)}f_{ik}((s_t^j,a_t^j)|j\in\Delta(k))r^k(s_t^k,\{a_t^j|j\in\Delta(k)\})\right]$$

$$\leq\sum_{k\in\mathcal{L}}\mathbb{E}_{\{\pi^j|j\in\Delta(k)\}}\left[\sum_{t=0}^{\infty}\gamma^t r^k(s_t^k,\{a_t^j|j\in\Delta(k)\})\right]$$

$$=\sum_{k\in\mathcal{L}}V_k^{\{\pi^j|j\in\Delta(k)\}}(s_0^k). \tag{14}$$

Finally, by combining (13) and (14), we derive

$$\sum_{i\in\mathcal{V}}\tilde{V}_i^{\{\pi^j|j\in\Omega(i)\}}(s_0^i)\leq\sum_{i\in\mathcal{L}}V_i^{\{\pi^j|j\in\Delta(i)\}}(s_0^i). \tag{15}$$

## D  OVERVIEW OF THE ALGORITHM

In this section, an overview of the proposed training algorithm for MARLM-SR, including the pseudo-code, is presented. The algorithm consists of the outer and inner settings as shown in Fig. 8. In Fig. 8(a), in the inner setting, the followers perform their subtasks in a DAG every time step. In Fig. 8(b), in the outer setting, two different types of agents are trained to guide the followers to achieve a high team reward. First, the leader provides goals to followers in each goal period based on the global state at the beginning of a goal period. After each goal period, reward generator and distributor (RGD) provides synthetic rewards to guide the followers to have a better exploration based on their achievements within the goal period. Here, a single agent plays both the roles of the reward generator and the reward distributor. Specifically, the reward generator generates a synthetic reward and the reward distributor distributes the produced synthetic reward from the highest level followers (sinks) to the lowest level followers following the DAG based on the achievement of the followers within the goal period. The team reward based on the follower's achievements in the episode is given to all outer agents. In other words, these three outer agents must learn the policy to produce and distribute the goals and the synthetic reward that coordinates the followers. On the other hand, in the inner setting, the followers learn policies for high team and synthetic rewards by following the given goals well. If the followers are guided well based on the policies of the outer agents and a high team reward is achieved, this high team reward is given not only to the followers but also to the outer agents.

In the algorithm, the leader sets goals at the beginning of each goal period and is rewarded after the goal period. On the other hand, the RGD determines the synthetic reward distribution strategy after each goal period. And this strategy influences the followers to behave differently in the next goal periods. Therefore, the RGD is rewarded in the next goal period.

## E  EXPERIMENTAL ENVIRONMENTS

We evaluated our model in diverse and challenging environments with DAG constraints including a real-world scheduling task. First, we created three environments to simulate DAG systems as shown in Fig. 9.

**Factory production planning case.** As shown in Fig. 9(a), we construct a DAG with four nodes and a depth of three in this case. In each episode, across ten goal periods with each lasting 40 steps, the objective of the four agents is to maximize revenue by producing final products in accordance with both the product's value and the demand. The agents must cooperate to achieve this objective, given

---

**Algorithm 1** The proposed MARLM-SR algorithm

---

**Require:** The number of steps for a goal period $D$, GSF step size $k$;
1: Initialize buffer replay $\mathcal{B}^i, i \in \mathcal{V}$ for each follower $i$ in $\mathcal{V}$, where $\mathcal{V}$ represents the node set of the DAG;
2: Initialize buffer replay $\mathcal{B}^L$ and $\mathcal{B}^{RGD}$ for the leader and the RGD;
3: **for** each episode **do**
4:     Reset the environment and observe the initial state $S_{01} = (s_{01}^i | i \in \mathcal{V})$;
5:     **for** each goal period $l$ **do**
6:         $gsf_l \leftarrow S_{l1}$;
7:         $S_l^L \leftarrow S_{l1} \circ (g_{l-1}^i | i \in \mathcal{V}) \circ (sr_{l-1}^i | i \in \mathcal{V})$;
8:         Produce the set of goals $(g_l^i | i \in \mathcal{V}) \sim \pi^L(\cdot | S_l^L)$;
9:         **for** $d = 1, 2, \cdots, D$ **do**
10:             Set team reward $R_{ld} \leftarrow 0$
11:             **for** each agent $i \in \mathcal{V}$ from root nodes to sinks **do**
12:                 Sample $a_{ld}^i = \pi^i(\cdot | s_{ld}^i)$;
13:                 Execute action $a_{ld}^i$;
14:                 **if** $d < D$ **then**
15:                     Observe new state $s_{l,d+1}^i \sim p(\cdot | s_{ld}^i, \{a_{ld}^j | j \in \Delta(i)\})$;
16:                 **else**
17:                     Observe new state $s_{l+1,1}^i \sim p(\cdot | s_{ld}^i, \{a_{ld}^j | j \in \Delta(i)\})$;
18:                 **end if**
19:                 **if** $i \in \mathcal{L}$, where $\mathcal{L}$ represents sink nodes in the DAG **then**
20:                     Receive reward $r_{ld}^i$;
21:                     $R_{ld} \leftarrow R_{ld} + r_{ld}^i$;
22:                 **end if**
23:             **end for**
24:             **if** $d \bmod k = 0$ **then**
25:                 $gsf_l \leftarrow gsf_l \circ S_{l,d+1}$;
26:             **else if** $d = D$ **then**
27:                 $gsf_l \leftarrow gsf_l \circ S_{l+1,1}$;
28:             **end if**
29:             **if** $d = D$ **then**
30:                 $S_l^{RGD} \leftarrow gsf_l \circ (g_l^i | i \in \mathcal{V})$;
31:                 Produce $(q_l) \circ (v_l^i | i \in \mathcal{V}) \circ (e_l^{(i,j)} | (i,j) \in \mathcal{A}) \sim \pi^{RGD}(\cdot | S_l^{RGD})$, where $\mathcal{A}$ represents the arc set of the DAG;
32:                 Compute $(sr_l^i | \forall i \in \mathcal{V})$ based on $(q_l)$, $(v_l^i | i \in \mathcal{V})$, and $(e_l^{(i,j)} | (i,j) \in \mathcal{A})$ using the proposed reward generation and distribution algorithm;
33:             **end if**
34:             **for** each agent $i \in \mathcal{V}$ **do**
35:                 **if** $d < D$ **then**
36:                     $cr_{ld}^i \leftarrow \frac{R_{ld}}{|\mathcal{V}|}$;
37:                 **else**
38:                     $cr_{ld}^i \leftarrow \frac{R_{ld}}{|\mathcal{V}|} + sr_l^i$;
39:                 **end if**
40:                 $\mathcal{B}^i \leftarrow \mathcal{B}^i \cup \{\{s_{ld}^i, a_{ld}^i, cr_{ld}^i\}\}$;
41:             **end for**
42:         **end for**
43:         $\mathcal{B}^L \leftarrow \mathcal{B}^L \cup \{\{S_l^L, (g_l^i | i \in \mathcal{V}), \sum_{d=1}^D R_{ld}\}\}$;
44:         **if** $l \neq 0$ **then**
45:             $\mathcal{B}^{RGD} \leftarrow \mathcal{B}^{RGD} \cup \{\{S_{l-1}^{RGD}, (q_{l-1}) \circ (v_{l-1}^i | i \in \mathcal{V}) \circ (e_{l-1}^{(i,j)} | (i,j) \in \mathcal{A}), \sum_{d=1}^D R_{ld}\}\}$;
46:         **end if**
47:     **end for**
48:     Update each $\pi^i$ for $i \in \mathcal{V}$, $\pi^L$, and $\pi^{RGD}$ using their respective buffers $\mathcal{B}^i$, $\mathcal{B}^L$, and $\mathcal{B}^{RGD}$;
49: **end for**

---

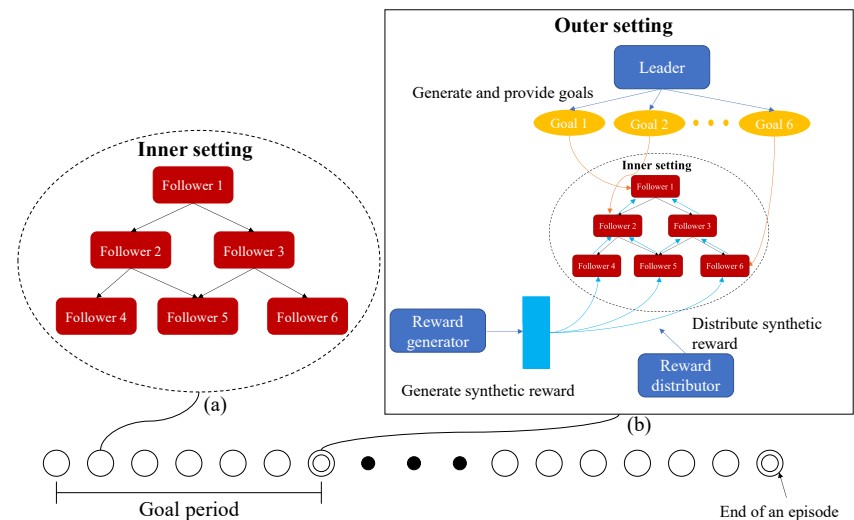

Figure 8: An overview of the proposed training algorithm.

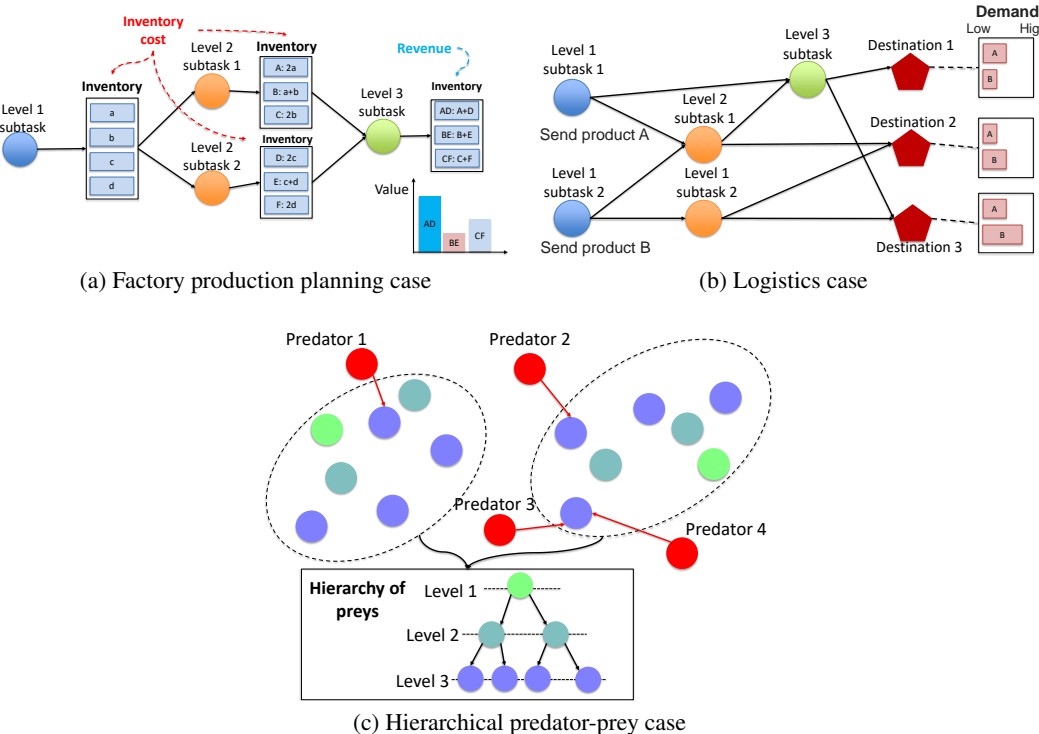

(a) Factory production planning case

(b) Logistics case

(c) Hierarchical predator-prey case

Figure 9: Illustrations of the use cases. Cooperation is necessary (a) to achieve a high profit, (b) to satisfy demands while minimizing costs, and (c) to avoid being caught by predators for as long as possible.

that the values of the final products change every 40 steps. Additionally, costs are incurred from both overproduction and parts that remain unused in the final products. In every goal period, each final product is assigned a value of 2, 3, or 4 uniformly at random with each final product having a distinct value. We have set these values to create a significant difference in the value of each product. For instance, the most valuable product holds a value twice that of the least valuable one. The agents should cooperate to prioritize the production of the most valuable product first. In addition, the three

products collectively share a total demand of 10 in each goal period, distributed randomly. Ten is the maximum number of final products that can be produced from scratch within a single goal period. We randomly set the product values and demand to reflect a dynamic production and market environment.

In each subtask, a machine may or may not produce one type of a product. If a machine chooses to produce a product, it consumes required parts, which are denoted by the blue boxes in the inventories, and stores the produced product in its own inventory. For example, if the machine for level 2 subtask 1 produces 'B' and stores it, it consumes 1 part 'a' and 1 part 'b.' A machine cannot produce a product without the necessary parts. They are subject to bill of material (except the machine in level 1). Final products are made after processing all three levels. Revenue is the cumulative value of the produced final products after an episode, and the factory is rewarded according to this revenue. Meanwhile, the factory is penalized by inventory holding costs of stored parts. Inventory holding costs of 0.3 and 0.8 are assigned to each part in the inventory of level 1 and level 2. Regarding overproduction, a penalty of 1 is imposed for each final product that is overproduced. We set two holding costs and the overproduction cost randomly, but the cost imposed at a higher level is set to be greater than that at a lower level.

**Logistics case.** In this case, the subtasks are to send a product either 'A' or 'B' at a time step. Similar to the factory case, each node must have a product in the inventory given from lower level nodes in order to send the product. Only the nodes in level 1 can send a product without constraint, but they can provide only one type of a product. Each node can choose not to send any product as well. If a product passes through all intermediate nodes in the graph, it arrives at one of the three destinations that have specific demands for both 'A' and 'B.' In summary, as shown in Fig. 9(b), we have a three-level DAG with five nodes, excluding the three destinations. This is because no tasks are required at these destinations.

Each episode consists of 300 time steps with 30 goal periods. In this case, we introduce randomness into both the demand for the two products at the destinations and the shipping cost on each arc to simulate a dynamic logistics scenario. The demand and the shipping cost vary in every epoch. Specifically, we assign a shipping cost, drawn from a uniform distribution [0, 0.3], to each arc in order to create subtle differences between them. Additionally, the demand follows the uniform distribution with the lower and upper bounds in Table 1. We categorize destinations into high, medium, and low demand to diversify demand based on the difficulty of reaching each one. For instance, to reach destination 1, a product needs to traverse 1.67 arcs, while destination 2 requires an average of 2.00 arcs, and destination 3 requires an average of 2.50 arcs. At each destination, benefits are granted if the demands are met. Specifically, benefits of 100, 300, and 200 are set for destinations 1, 2, and 3, respectively. However, additional inventory holding cost or inventory shortage cost is given if the demand is not met at each destination, and inventory holding cost is imposed at each node in the graph as well. The inventory holding cost at levels 1, 2, and 3 is 0.3. We set the maximum shipping cost and the inventory holding cost at each node to 0.3. This allows them to impact the total benefit, but not significantly. Meanwhile, we set the additional inventory holding cost at the destinations to 3, and the inventory shortage cost at the destinations to 8, so that they significantly influence the total benefits. The inventory shortage cost is set higher than the inventory holding cost, which encourages agents to prioritize shipping over holding inventory. All costs are deducted from the total benefits to calculate the team rewards. Consequently, the maximum reward amounts to 600.

Table 1: The lower and upper bounds of the uniform distribution for each product at the three destinations in the logistics case

| Destination | Product | Lower bound | Upper bound |
|---|---|---|---|
| Destination 1 | A | 5 | 10 |
| (Low demand) | B | 3 | 7 |
| Destination 2 | A | 110 | 130 |
| (High demand) | B | 70 | 90 |
| Destination 3 | A | 35 | 45 |
| (Medium demand) | B | 80 | 100 |

**Hierarchical predator-prey case.** In this variant of the classic predator-prey game, the preys have a hierarchy in which a higher level prey follows the parent prey (refer to the structure of the DAG in Fig. 9(c)). The objective of this game is to guide the prey at the sink nodes to evade predators and

survive for as long as possible. Each time predators move one step to chase the preys in the highest level. The preys also can move one step to move away from the predators. A prey can choose to move in any direction, but cannot go beyond the boundary set by the parent prey. The boundary is 5 steps away from the parent prey. The total survival time of the preys at the sink nodes is given as the team reward after each episode. In other words, the later the preys in the highest level are caught, the higher rewards are given to the set of the preys. Thus, the parent preys must guide their children well to provide safe areas to the preys in the highest level. An episode concludes when all the prey at the sink nodes are caught, or when the maximum duration of 200 steps is reached. We set the length of the goal period to 10. In each episode, the predators and the preys start at fixed positions, but the predators randomly select their direction.

**Real-world production planning case.** We also investigated the performance of the proposed algorithm in a real-world scheduling environment. For this environment, we developed a simulator that sets a reasonable demand goal for each product type within a given time period based on the actual production information of one of Intel's high volume packaging and test factory (AT factory). The purpose of this task is to schedule jobs so that each job goes through a specific sequence of operations defined by product type. In this scheduling environment, the precedence constraints of operations are not different according to product type, even though each product type requires a different set of operations. For example, if operation 'b' comes after operation 'a' for a product 'A,' 'b' cannot come earlier than 'a' for any product. Thus, the unique DAG is built based on these precedence constraints. Several stations are assigned for an operation and one station can only process one operation. Thus, we group stations by target operation, and each group of stations is called a station family. Even members in the same station family have a different set of products available. In addition, the processing time of a product significantly varies depending on the type of operation. In summary, jobs must be scheduled to meet demand goals while considering all these operational constraints and relationships between operations. The AT factory is a large-scale line that contains more than 75 stations, 10 operations, and 35 product types. This scheduling task is very challenging.

In this study, we simulated an environment where jobs need to be scheduled for five shifts, with each shift lasting 12 hours (refer to Fig. 10). In the figure, only nine product types are present (right vertical legend). The left vertical axis corresponds to stations. It shows that not all products are processed in the same shift, but the AT factory has to fabricate a different set of products each shift to meet ever-changing demands. In addition, the AT factory has a strict constraint that for most station families only one product conversion is allowed in a shift. In other words, with the exception of a few families that can execute multiple conversions, the other families have only one opportunity to select a station and change product type for the selected station in a shift. Thus, it is catastrophic if an agent makes a wrong conversion decision because the waiting jobs that are not current setup will not have a chance to be processed for a long time. Therefore, an agent makes a conversion decision for the assigned station family, including which station will perform the conversion and what the next product type will be.

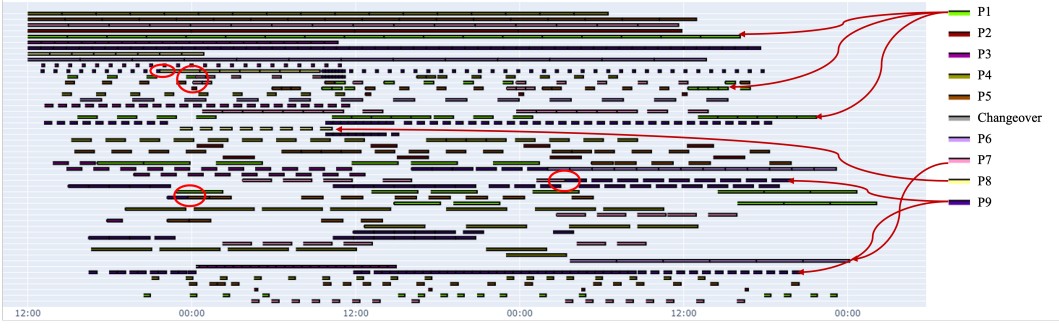

Figure 10: An example Gantt chart for a five shift schedule. We use arrows to distinguish a few products, and we highlight some changeovers with a red circle.

## F Implementation details

We implemented the proposed algorithm and the comparison algorithms by using the proximal policy optimization (PPO) algorithm as the baseline learning algorithm for each agent in all MARL algorithms. A fully-connected neural network with 2 layers of size 256 with ReLU activations is applied to both actor and critic of each agent. Entropy regularization Schulman et al. (2017a) is applied with a coefficient of 0.01. We summarize the other hyperparameters for RL in Table 2. For the three artificial benchmark cases, we set the global state flow (GSF) $gsf_l$ at the $l$-th goal period to include six global states for the factory production planning and logistics cases and ten global states for the hierarchical predator-prey case, each with equal intervals. For the sensitivity analysis against the length of the goal period, the number of included global states is fixed. In the real-world production planning case, we include only the initial and final global state of each goal period in the GSF. The length of the goal vector $g_l^i$, where $i \in \mathcal{V}$, is set to 3.

Table 2: Hyperparameters for our model and the baselines.

| Hyperparameter | Value |
|---|---|
| Episode length | 1,200 |
| Batch size | 256 |
| Learning rate | 0.0001 |
| Discount factor | 0.9900 |
| Clipping value | 0.2000 |
| Generalized advantage estimation parameter lambda | 0.9500 |

In our implementation, we included only the initial global state $S_{l1}$ at each goal period $l$ as the input state vector for the leader, to enhance tractability. For RGD, we introduced two separate networks using the same state vector; one for the reward generator that produces $q_l$, and another for the reward distributor that creates $(v_l^i | i \in \mathcal{V})$, $(e_l^{(i,j)} | (i,j) \in \mathcal{A})$, and finally generates synthetic rewards. To obtain synthetic rewards, we first need to calculate the average team reward per goal period, $\frac{\overline{R_e}}{N_e}$. In experiments, we only considered the number of goal periods and the total team reward from the immediately preceding episode. All algorithms compared in this work were implemented based on the TensorFlow framework.

## G Ablation analysis in scheduling scenarios

We compared the five baselines for the ablation study in diverse scheduling scenarios. Specifically, we trained the agents in the DAG using the five baseline algorithm: GS, SRM, LFM, RFM, and the proposed algorithm, and simulate 1,000 scheduling episodes using the trained models. Fig. 11 shows the histogram of the completion rate on 1,000 episodes for each baseline, where each baseline has the same x-axis values. Here, the completion rate is the ratio of the lots that pass all required operations to demanded lots. We also conducted statistical significance tests to validate whether the proposed algorithm is significantly better than the other baselines in terms of the completion rate as shown in Table 3. In the figure and the table, we do not report the average value of the completion rate and the range of the histogram for confidentiality. Instead, the table reports the average improvement in the completion rate compared to GS. First, by introducing the concept of MARL, we were able to improve the mean completion rate by approximately 150.0%. This component contributes most significantly to the performance improvement. A comparison between SRM and LFM/RFM reveals that the leader and the RGD also contribute to performance improvement. Specifically, the RGD improves the mean completion rate by 3.9%. Furthermore, by introducing both outer agents, we are able to improve the mean completion rate by 8.5%. As a result, the proposed algorithm demonstrates a higher overall completion rate compared to other baselines. It achieves approximately a 170.0% improvement in the mean completion rate compared to GS.

## H Details of sensitivity analyses

We conducted sensitivity analyses to validate the effect of the goal period length using the three artificial benchmark cases. We established four length levels: short, medium, long, and extremely

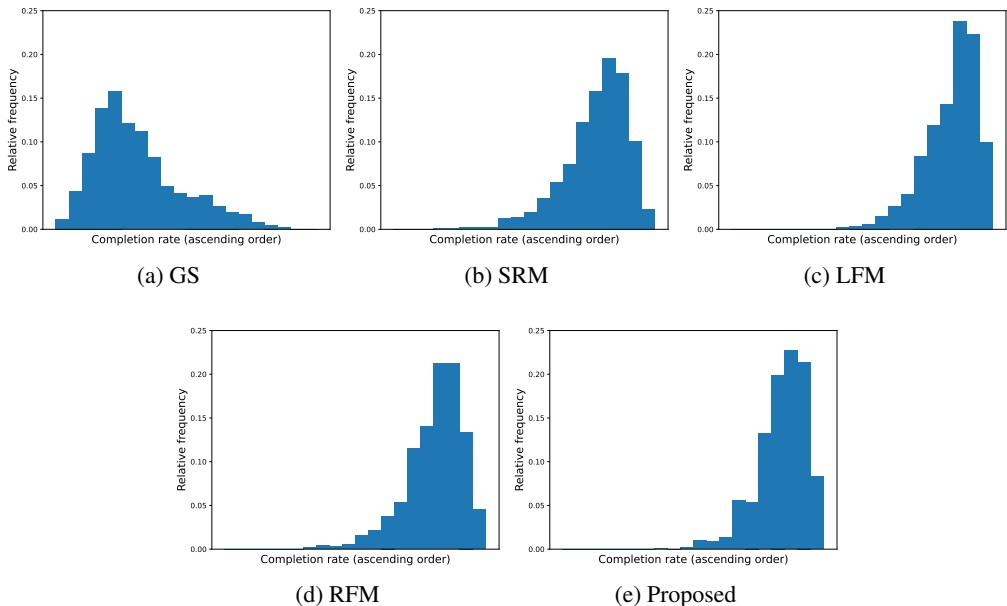

Figure 11: The histogram of completion rate over 1,000 scheduling scenarios (episodes) for the ablation study.

Table 3: Result of statistical significance tests.

|  | GS | SRM | LFM | RFM | Proposed |
|---|---|---|---|---|---|
| Mean improvement | - | 0.464 | 0.529 | 0.494 | 0.529 |
| p-value | <0.001 | <0.001 | 0.555 | <0.001 | - |

long (ext. long). These were determined considering the maximum episode length for each case, as shown in Table 4. The results are presented in Fig. 12, where we applied a moving window technique with a window size of 100 and a step size of 1 in displaying the results.

Table 4: Details of the sensitivity analyses setting for the length of goal period.

| Case | Short | Medium | Long | Ext. long | Episode length |
|---|---|---|---|---|---|
| Factory production planning | 20 | 40 | 100 | 200 | 400 |
| Logistics | 5 | 10 | 30 | 60 | 300 |
| Hierarchical predator-prey | 10 | 20 | 50 | 100 | 200 |

In general, a longer goal period results in poorer performance. For instance, an extremely long goal period yields a performance drop of 78.4%, 58.1%, and 51.3% compared to the best-performing lengths for factory production planning, logistics, and hierarchical predator-prey cases, respectively. This performance comparison is based on the average team reward over the last 10,000 episodes. However, a short goal period does not always ensure optimal performance. Specifically, we achieved a team reward improvement of 8.8% and 21.4% with a medium goal period length compared to a short goal period for factory production planning and logistics cases, respectively. Thus, the goal period length should be optimized based on the environment's characteristics.

We also analyzed sensitivity regarding the dimension of the goal vector produced by the leader. Fig. 13 illustrates that we can generally achieve performance improvements by increasing the dimension of the goal vector. Specifically, a 44.2% performance improvement is observed when increasing the dimension from 1 to 3. However, the magnitude of improvement significantly decreases as the dimension increases; only an 11.1% improvement is obtained when setting the dimension to 5. Even though we observed this improvement in the factory production planning case (23.9%), the

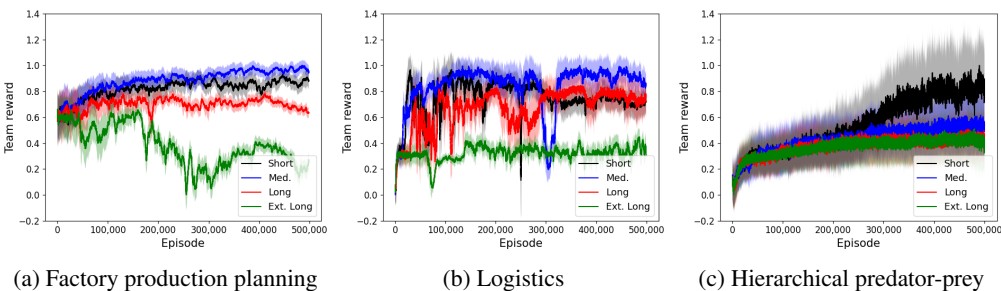

(a) Factory production planning    (b) Logistics    (c) Hierarchical predator-prey

Figure 12: Sensitivity analyses results on the three benchmark cases. Min-max normalization is applied to the team reward to standardize the scale of the y-axis across the three cases.

team reward improvement is not statistically significant in the hierarchical predator-prey scenario, with the p-value higher than 0.1 in a two-sample t-test. In summary, significant performance gains are achievable only when the goal vector has a very limited dimension. High dimension increases computation time due to the enlarged state space for all followers.

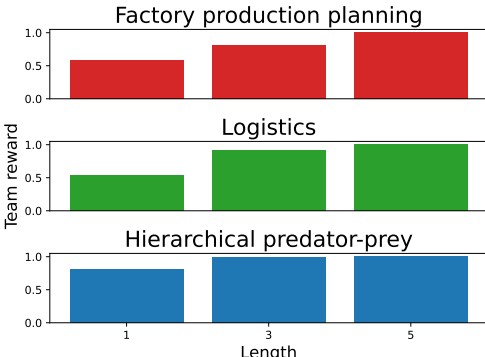

Figure 13: Results of sensitivity analysis on the dimension of the goal vector generated by the leader, measured using the average team reward over the last 10,000 episodes during training.

