# OpenReview forum: "Learning Multiple Coordinated Agents under Directed Acyclic Graph Constraints"
_ICLR.cc/2024/Conference — Submitted to ICLR 2024_

### Official Review · Reviewer_e4e1 · 2023-10-29

**Soundness:** 2 fair
**Presentation:** 1 poor
**Contribution:** 2 fair
**Rating:** 3
**Confidence:** 3

**Summary:**

This paper introduces a Multi-Agent Reinforcement Learning (MARL) approach designed to train multiple coordinated agents, taking into account the constraints of directed acyclic graphs (DAG). The highlighted features of this solution include:

1. From a computational standpoint, the authors leverage the inherent properties of DAG structures to enhance the solution's efficiency. This is realized by introducing innovative concepts such as the leader agent, reward generator, and distributor agent. These elements guide the subordinate follower agents in a more targeted exploration of the parameter space in environments governed by DAG constraints.

2. On a theoretical level, the proposed solution serves as a lower bound for the optimal value function. This is made possible by incorporating a surrogate value function supplemented with synthetic rewards.

Furthermore, the authors validate the efficacy of their method on four distinct DAG environments, underscoring its advantages in comparison to non-DAG strategies.

**Strengths:**

- This paper introduces an approach to address multi-agent coordination challenges within the framework of directed acyclic graphs (DAG), a context particularly relevant to real-world scenarios like industrial process control.

- The solution proposed is very interesting, with a special emphasis on the reward generator. This component effectively shapes rewards at the multi-agent coordination level, aligning with the specificities of DAG constraints.

- From a theoretical standpoint, the method appears robust and well-founded.

- To underscore the effectiveness of their approach, the authors have carried out comprehensive experiments on tasks under DAG constraints.

**Weaknesses:**

- The motivation of this work, as it stands, could be better articulated. While I recognize that problems with multiple subtasks are important for multi-agent coordination, it's not clearly conveyed why these specifically fall under multi-agent reinforcement learning or what challenges they present. Given that the work's contribution is centered on the DAG structure, it might be beneficial for the authors to use an illustrative example for a clearer understanding.

- In line with the above point, I find it somewhat challenging to connect the proposed solution with the problem described. For instance, the leader is trained to generate goal vectors for followers, but how does this align with the DAG constraint of the problem? How can we determine that a proposed goal-vector results in "better coordination"? A clearer outline of the challenges and more detailed motivations for the solution would be helpful.

**Questions:**

1. If different agents deal with different subtasks within the DAG, then do there exist interactions between agents in this setting?

2. Is it possible to directly derive the optimal value function for the DAG? If not, what impediments exist in directly computing this function? Might it be feasible in a more simple setup, such as in a tabular setting?

3. The proposed approach maintains "abstract" messages between the leader agent and its followers. Why this level of abstraction? Could this characteristic of non-interpretability pose challenges to the method's practical application?

4. With respect to the synthetic reward, how is an agent's contribution to the team reward assessed quantitatively? If a particular subtask significantly influences the coordination task but has several preceding subtasks, are these ancestors also assigned with high synthetic rewards?

5. The study appears to incorporate several assumptions:
a)	In P3: “the team reward is the sum of the rewards obtained from sinks”
b)	In P4: “there can be a function $f_{ik}$ that measures the contribution of agent $i$ to sink agent $k$’s reward and …”
c)	In P6: “the synthetic reward for the follower $i$ is determined based on its contributions to the sink followers among its descendants”

Do these assumptions still hold in real-world scenarios? Might they limit the broader applicability of the proposed method? A more in-depth discussion on this matter would be appreciated.

---

> ### Author Response · Authors · 2023-11-20
>
> We appreciate your advice and comments. We believe this rebuttal process has helped us to further develop our work into a substantially better manuscript, and we sincerely appreciate the opportunity to do so. We have modified all the required sections according to your comments and provided responses one by one.
>
> *Weakness 1: The motivation of this work, as it stands, could be better articulated. While I recognize that problems with multiple subtasks are important for multi-agent coordination, it's not clearly conveyed why these specifically fall under multi-agent reinforcement learning or what challenges they present. Given that the work's contribution is centered on the DAG structure, it might be beneficial for the authors to use an illustrative example for a clearer understanding.*
>
> - In our study, we applied the proposed algorithm to a real-world scheduling scenario in Intel’s high-volume packaging and test factory after modeling the factory as a DAG based on the precedence constraints of subtasks. This use case instigated this research. Considering the semiconductor industry's market size (forecasted to be $625 billion in 2023 and $1,307 billion in 2032, as per https://market.us/report/semiconductor-market/), it is important to address the scheduling challenges in semiconductor manufacturing (and other high-tech industries) using a DAG-based approach. Furthermore, many IoT applications in cyber-physical systems, such as smart grids, require control actions for multiple agents following a DAG structure (Khare et al., 2019). With industry systems becoming increasingly complex, the DAG setting is likely to become more prevalent in real-world scenarios.
>
>   In many real-world DAG environments, information about state transitions and reward functions is unknown, especially considering the complex relationships among multiple subtasks. For addressing this model-free control problem in MDP-DAG settings, we believe that multi-agent reinforcement learning can yield the most effective strategy.
>
>   We have created a ‘Motivation’ section in our Appendix to highlight the importance of DAG setting and added an illustrative example for a clearer understanding.
>
> Khare, S. et al (2019). Linearize, predict and place: minimizing the makespan for edge-based stream processing of directed acyclic graphs. In *Proceedings of the 4th ACM/IEEE Symposium on Edge Computing*.
>
> *Weakness 2: In line with the above point, I find it somewhat challenging to connect the proposed solution with the problem described. For instance, the leader is trained to generate goal vectors for followers, but how does this align with the DAG constraint of the problem? How can we determine that a proposed goal-vector results in "better coordination"? A clearer outline of the challenges and more detailed motivations for the solution would be helpful.*
>
> - In an environment characterized by numerous subtasks with complex interrelations, each agent, having only partial observability, needs to ascertain the system's overall status toward a high team reward. In such cases, a leader can direct each agent by providing an abstract goal that is trained to achieve large team reward.
>
> - The leader is introduced to support the RGD, as the RGD alone cannot adequately guide the followers. The experimental results demonstrate that the introduction of the leader is essential. This implies that the proposed goal vectors lead to better coordination, as we assume that better coordination yields higher team rewards.
>
> *Question 1: If different agents deal with different subtasks within the DAG, then do there exist interactions between agents in this setting?*
>
> - Yes, there can be interactions even if agents are dealing with different subtasks. The arcs in the DAG represent these interactions.
>
> *Question 2: Is it possible to directly derive the optimal value function for the DAG? If not, what impediments exist in directly computing this function? Might it be feasible in a more simple setup, such as in a tabular setting?*
>
> - If the reward functions are known for all agents in an MDP-DAG, taking into account the directional relationships defined by the DAG, we can directly derive the optimal value function in a simple setup. However, reward functions of agents do not exist and thus are unknown.

---

> ### Author Response · Authors · 2023-11-20
>
> *Question 3: The proposed approach maintains "abstract" messages between the leader agent and its followers. Why this level of abstraction? Could this characteristic of non-interpretability pose challenges to the method's practical application?*
>
> - To define interpretable goals (messages) requires deep domain knowledge. Additionally, a goal defined with strong background knowledge may not ensure better coordination. This is the reason why we introduce a non-interpretable goal. However, the characteristic of non-interpretability poses a challenge to acquiring useful strategies for coordinating subtasks through learning.
>
>   We believe that comparing interpretable and non-interpretable goals within the same multi-agent environment would be beneficial. Due to time constraints, we are unable to design and implement an interpretable goal for comparison during this discussion period. Therefore, we kindly request the opportunity to include such results in the camera-ready version.
>
> *Question 4: With respect to the synthetic reward, how is an agent's contribution to the team reward assessed quantitatively? If a particular subtask significantly influences the coordination task but has several preceding subtasks, are these ancestors also assigned with high synthetic rewards?*
>
> - Our concept is that a high team reward is achieved if the synthetic reward is distributed effectively, reflecting an accurate assessment of each agent's contribution. However, we cannot quantify this contribution. Instead, we expect the RGD agent to assess the followers correctly and achieve high rewards through learning.
>
>   A large contribution from an agent does not necessarily mean that preceding subtasks have a large contribution. The role of the RGD agent is to capture the individual contributions of each agent within the system.
>
> *Question 5: The study appears to incorporate several assumptions: a) In P3: “the team reward is the sum of the rewards obtained from sinks” b) In P4: “there can be a function that measures the contribution of agent to sink agent ’s reward and …” c) In P6: “the synthetic reward for the follower  is determined based on its contributions to the sink followers among its descendants. Do these assumptions still hold in real-world scenarios? Might they limit the broader applicability of the proposed method? A more in-depth discussion on this matter would be appreciated.*
>
> - First, 'the team reward is the sum of the rewards obtained from sinks’ applies to most DAG systems, particularly in industrial processes, because the achievement of a system can be evaluated only after all subtasks in DAG systems are completed. The other two assumptions have been introduced exclusively for the theoretical analysis. They are not used in the model and methodology. Therefore, we believe that these three assumptions do not restrict the applicability of our proposed algorithm.

---

> > ### Comment · Reviewer_e4e1 · 2023-11-21
> >
> > Thank you very much for your response!
> >
> > In terms of the motivation behind this study, I concur with the authors on the significant influence of the semiconductor industry. However, my primary concern is that the paper does not distinctly delineate the challenges that DAGs present in this context. The authors have mentioned that in numerous DAG-based scenarios, information regarding state transitions and the reward function remains unknown. However, this description is rather vague and could apply to many other non-DAG-oriented environments. It's also unclear whether this issue is prevalent in all DAG-based environments, making it difficult to determine the applicability of this solution. If the paper's objective is to propose a universal solution, then the concept of a centralized leader is not novel [1]. If this is an industry-focused study, the authors should more explicitly outline the challenges specific to this domain.
> >
> > [1] M^3RL: Mind-aware Multi-agent Management Reinforcement Learning https://openreview.net/pdf?id=BkzeUiRcY7

---

> ### Author Response · Authors · 2023-11-22
>
> Thank you for your valuable comment. Based on your feedback, we have revised our paper to include additional challenges presented by DAGs. Please refer to the responses below for further details.
>
> - In non-DAG environments, agents perform subtasks independently, or they are only weakly connected without any precedence constraints, such that the result of each subtask directly affects the team's rewards. However, in DAG environments, a high-level agent is highly dependent on the results of lower-level agents (ancestors). As a result, the state and action space of a high-level agent are significantly affected by its ancestors. In particular, in the perspective of a low-level agent, the system does not receive a reward unless all its downstream agents have taken actions. Such a delayed rewarding mechanism is common in many real-world problems including industrial process control, traffic optimization, and resource allocation. Most existing deep reinforcement learning algorithms suffer from inferior performance because no immediate supervision is given. Furthermore, a low-level agent cannot directly affect the team's reward, but the team reward depends not only on this agent but also on its descendants. In summary, in DAG environments, it is crucial to consider these complex interrelationships as defined by a DAG.
> - We have added this content in the third paragraph of the Introduction.
> - Furthermore, we have detailed challenges associated with the example (Fig. 7) from our 'Motivation' section in the Appendix. For more information, please refer to the second paragraph of the motivation section.

---

### Official Review · Reviewer_DooC · 2023-10-29

**Soundness:** 3 good
**Presentation:** 3 good
**Contribution:** 3 good
**Rating:** 6
**Confidence:** 3

**Summary:**

This paper considers the problem of a Markov decision process with DAG constraints, which is prevalent in some real-world problems. To deal with the issues of delayed rewards and lack of individual incentives, the paper proposes the idea of synthetic rewards, which are based on each agents' contributions to the down-streaming cooperative tasks. Theoretically, it proves that the surrogate value function based on the synthetic rewards is the lower bound of the  optimal value function. Empirically, the paper proposes a training algorithm to generate the synthetic rewards. The proposed algorithm is tested on several MARL environments.

**Strengths:**

- The setting is very interesting. Many practical problems have inter-dependent subtasks which can be framed by the proposed MDP with DAG constraints.
- The idea of synthetic rewards is well motivated, since the team reward cannot well capture the individual contribution. Further, it is theoretically justified by the paper.
- A practical algorithm to generate the synthetic rewards is proposed and has achieved better performance on several MARL environments.

**Weaknesses:**

- The DAG constraints are predetermined, which may need some domain-knowledge and human annotations.
- Compared to decentralized algorithms such as independent Q-learning, the proposed practical algorithm requires a centralizer to generate the synthetic rewards, which may not always be available.
- In the practical algorithm, the goals generated by the leader are not interpretable. It is not clear why it is beneficial for the MARL problems.

**Questions:**

- There are no error bars in the Figure 3 and 5. Are the experiments tested on multiple seeds?
- Can the authors visualize the generated synthetic rewards and how does it captures the individal contribution? For example, when does the RGD generate low rewards and high rewards in the provided testing environments?
- The purpose of the goals generated by the leader is not clear to me. Can the authors explain why it is beneficial?

---

> ### Author Response · Authors · 2023-11-20
>
> We appreciate your advice and comments. We believe this rebuttal process has helped us to further develop our work into a substantially better manuscript, and we sincerely appreciate the opportunity to do so. We have modified all the required sections according to your comments and provided responses one by one.
>
> *Weakness 1: The DAG constraints are predetermined, which may need some domain-knowledge and human annotations.*
>
> - All other centralized and decentralized algorithms also require the knowledge of DAG constraints for implementation because what is observed are actions for the subtasks, and the actions are taken under the given DAG constraints. Thus, we argue that this a priori information requirement applies to reinforcement learning itself when it is used for an environment under DAG constraints.
>
> *Weakness 2: Compared to decentralized algorithms such as independent Q-learning, the proposed practical algorithm requires a centralizer to generate the synthetic rewards, which may not always be available.*
>
> - The centralizer is only an algorithmic entity/concept with no physical meaning. It necessitates accommodating predetermined DAG constraints. However, it is important to note that decentralized algorithms also require predetermined DAG constraints for their implementation. In this regard, our proposed algorithm can be implemented under the same conditions as decentralized algorithms.
>
> *Weakness 3 and Question 3: In the practical algorithm, the goals generated by the leader are not interpretable. It is not clear why it is beneficial for the MARL problems. The purpose of the goals generated by the leader is not clear to me. Can the authors explain why it is beneficial?*
>
> - In an environment characterized by numerous subtasks with complex interrelations, each agent, having only partial observability, needs to ascertain the system's overall status toward a high team reward. In such cases, a leader can direct each agent by providing an abstract goal that is trained to achieve a greater team reward.
>
>   To define an interpretable goal (message) requires deep domain knowledge. Additionally, a goal defined with strong background knowledge may not ensure better coordination. This is the reason why we introduce a non-interpretable goal.
>
>   Nonetheless, we believe that comparing interpretable and non-interpretable goals within the same multi-agent environment would be beneficial. Due to time constraints, we were unable to design and implement an interpretable goal for comparison during this discussion period. Therefore, we kindly request the opportunity to include such results in the camera-ready version.
>
> *Question 1: There are no error bars in the Figure 3 and 5. Are the experiments tested on multiple seeds?*
>
> - Each model was trained on approximately 500,000 episodes where each episode has a different initial environment, and the training results are averaged over 100 episodes. We have added shaded areas in Figures 3, 5, and 12 to represent standard deviation.
>
> *Question 2: Can the authors visualize the generated synthetic rewards and how does it captures the individal contribution? For example, when does the RGD generate low rewards and high rewards in the provided testing environments?*
>
> - Synthetic rewards are continually adjusted by the RGD to achieve higher team rewards, which in turn are also awarded to the RGD. Consequently, the RGD must capture the individual contributions. However, it remains unclear whether the RGD generates an appropriate amount of synthetic reward that accurately captures each agent's contribution. It is a great suggestion to visualize rewards and contributions. We started this task but we will not be able to finish it before the rebuttal period is over. They will be included in the camera-ready version.

---

> > ### Comment · Reviewer_DooC · 2023-11-22
> > **Reply to rebuttal**
> >
> > Thank you for the rebuttal. Most of my concerns have been addressed. I decide to keep my score.

---

### Official Review · Reviewer_cDFa · 2023-11-01

**Soundness:** 3 good
**Presentation:** 3 good
**Contribution:** 3 good
**Rating:** 3
**Confidence:** 4

**Summary:**

This paper addresses the challenges of Multi-agent Reinforcement Learning (MARL) in scenarios with complex subtask dependencies, represented by Directed Acyclic Graphs (DAGs). To tackle the issues of delayed rewards and reward distribution among agents, the authors introduce a novel algorithm featuring a "leader" that generates abstract goals for agents, and a "Reward Generator and Distributor (RGD)" to coordinate agents based on their contributions. The approach aims to enhance agent coordination and optimize their performance in intricate, real-world applications.

**Strengths:**

This paper studied a novel problem that hasn't been considered before. The idea is very novel. The empirical results show that their method could be efficient.

**Weaknesses:**

1. The writing could be improved. Currently, the problem-setting section is very confusing.

2. There is a GitHub url in this paper, which violates the double-blind principle.

**Questions:**

1. Please further decertify the problem-setting.
2. Is this DAG setting common in some real-world scenarios?

---

> ### Author Response · Authors · 2023-11-20
>
> We appreciate your advice and comments. We believe this rebuttal process has helped us to further develop our work into a substantially better manuscript, and we sincerely appreciate the opportunity to do so. We have modified all the required sections according to your comments and provided responses one by one.
>
> *Weakness 1 and question 1: The writing could be improved. Currently, the problem-setting section is very confusing. Please further decertify the problem-setting.*
>
> - To enhance comprehension, we now provide an illustrative example for the modeling setting in Appendix B.
>
> *Weakness 2: There is a GitHub url in this paper, which violates the double-blind principle.*
>
> - We have included a GitHub URL for reproducibility. According to the ICLR 2024 Author Guide, it is stated that ‘All supplementary code must be self-contained and zipped into a single file or can be downloaded via an anonymous URL.’ Since there is no information in the given URL that can be used to identify the authors and the repository is standalone, we believe it does not violate the double-blind principle.
>
> *Question 2: Is this DAG setting common in some real-world scenarios?*
>
> - In our study, we applied the proposed algorithm to a real-world scheduling scenario in Intel’s high-volume packaging and test factory after modeling the factory as a DAG based on the precedence constraints of subtasks. Considering the semiconductor industry's market size (forecasted to be \\$625 billion in 2023 and \\$1,307 billion in 2032, as per https://market.us/report/semiconductor-market/), it is important to address the scheduling challenges in semiconductor manufacturing (and other high-tech industries) using a DAG-based approach. Furthermore, many IoT applications in cyber-physical systems, such as smart grids, require control actions for multiple agents following a DAG structure (Khare et al., 2019). With industry systems becoming increasingly complex, the DAG setting is likely to become more prevalent in real-world scenarios.
>
> - We have created a ‘Motivation’ section in our Appendix to highlight the importance of the DAG setting.
>
> Khare, S. et al (2019). Linearize, predict and place: minimizing the makespan for edge-based stream processing of directed acyclic graphs. In *Proceedings of the 4th ACM/IEEE Symposium on Edge Computing*.

---

### Official Review · Reviewer_qXdU · 2023-11-03

**Soundness:** 3 good
**Presentation:** 3 good
**Contribution:** 3 good
**Rating:** 6
**Confidence:** 3

**Summary:**

This paper proposes a novel framework to solve the MDP-DAG problem with synthetic rewards as the rewards of the agent itself and its contribution to its descendants. Two new learning components, the leader agent and the reward generator and distributor, are introduced to solve the problem. The proposed method shows improved performance on multiple different tasks compared to several baselines.

**Strengths:**

The paper is well-written and the contributions are novel. The motivation behind the proposed learning components is clear. Decomposing agents' contributions to team rewards makes the inner learning simpler. The experiment results are comprehensive, with various environments and baselines.

**Weaknesses:**

As mentioned in the discussion, theoretical analysis is missing for the convergence and optimality of the learned reward distributor.

The proposed method is very complex multiple inner and outer cycles, as well as additional learning components. May need more results or analysis on the stability and sensitivity to hyper-parameters of the proposed method.

**Questions:**

Can more results or analysis on the stability and sensitivity to hyper-parameters of the proposed method, in addition to the analysis of the goal period length in Figure 6?

---

> ### Author Response · Authors · 2023-11-20
>
> We appreciate your advice and comments. We believe this rebuttal process has helped us to further develop our work into a substantially better manuscript, and we sincerely appreciate the opportunity to do so. We have modified all the required sections according to your comments and provided responses one by one.
>
> *Question 1: Can more results or analysis on the stability and sensitivity to hyper-parameters of the proposed method, in addition to the analysis of the goal period length in Figure 6?*
>
> - We have conducted an additional sensitivity analysis on the dimension of the goal vector. We found that significant performance gains are achieved by increasing the dimension, but only when the goal vector has a very limited dimension.
>
>   We have incorporated the details of this analysis to Section H of Appendix.

---

### Meta-Review · Area_Chair_XJwf · 2023-12-10

**Metareview:**

The paper proposes a novel framework to address multi-agent reinforcement learning challenges in environments with complex subtask dependencies represented as directed acyclic graphs. It introduces new components like a leader agent and reward generator/distributor to provide synthetic rewards based on agents' contributions.

The strength is the novelty of the idea. Meanwhile, some weaknesses identified are the unclear motivation, lack of analysis on convergence/optimality of the reward distributor, complexity of the approach, and need for more sensitivity analysis. In the response, the authors have addressed questions on sensitivity analysis, DAG applicability, interaction between agents, and interpretability of the leader's goals.

Overall, the core idea is promising. However, I agree with reviewer e4e1 that the motivation may not be sound enough, weakening the contribution's significance. I moreover doubt that, from the scenario examples, DAG may not be sufficient to describe the real-world constraints.

**Justification For Why Not Higher Score:**

The core contribution is a new class of multiagent problems with DAG constraints. However, it is doubted whether DAG constraints bring significant challenges and whether DAG is a practical way to describe the real-world constraints.

**Justification For Why Not Lower Score:**

N/A

---

### Decision · Program_Chairs · 2024-01-16

Reject